# High-Gain Millimeter-Wave Beam Scanning Transmitarray Antenna

**DOI:** 10.3390/s23104709

**Published:** 2023-05-12

**Authors:** Hao-Zheng Yang, Shi-Wei Qu

**Affiliations:** School of Electronic Science and Engineering, University of Electronic Science and Technology of China, Chengdu 611731, China

**Keywords:** high gain, beam scanning, transmitarray antenna, millimeter-wave band

## Abstract

In this article, a high-gain millimeter-wave transmitarray antenna (TAA) maintaining scanning ability is developed, integrating an array feed as the primary emitter. The work is achieved within a limited aperture area, avoiding the replacement or extension of the array. The addition of a set of defocused phases along the scanning direction to the phase distribution of the monofocal lens allows the converging energy to be dispersed into the scanning scope. The beam forming algorithm proposed in this article can determine the excitation coefficients of the array feed source, and is beneficial to improve the scanning capability in array-fed transmitarray antennas. A transmitarray based on the square waveguide element illuminated by an array feed is designed with a focal-to-diameter ratio (F/D) of 0.6. A 1-D scan with a scope of −5° to 5° is realized through calculation. The measured results show that the transmitarray can achieve a high gain, 37.95 dBi at 160 GHz, although a maximum 2.2 dB error appears compared with the calculation in the operating band of 150–170 GHz. The proposed transmitarray has been proven to generate scannable high-gain beams in the millimeter-wave band and is expected to demonstrate its potential in other applications.

## 1. Introduction

The exploration of the millimeter-wave (mm-wave) spectrum opens growing application scenarios for future communication systems, including occasions where high-gain antennas and beam-steering antennas are required in the meantime. Traditional technical solutions for achieving high-gain beam scanning include phased array antennas (PAAs), reflective surface antennas and lens antennas. Although PAA technology has developed rapidly in recent years, the method of forming a very narrow scanning beam directly through radiation requires a large number of active channels, so its development is still restricted by the complex design and high cost in high-gain beam scanning applications. Reflector surface antennas, although built at low cost, suffer from inherent drawbacks such as feed source masking and relatively small scanning range. Transmitarray antennas constitute a solution to high gain antennas, which attracts growing research interests in recent years. To address the aforementioned issue, this article proposes an array-fed beam scanning planar TAA. Using the focusing effect of the planar lens, the electric large radiation aperture, which requires a large number of sampling points, is transformed to a smaller scale sampling surface, and then the array feed is used for sampling, thus significantly reducing the number of sampling points, i.e., the number of active channels, and achieving low-cost and high-gain beam scanning.

When it comes to beam-scanning antenna schemes, there are various candidates, such as phased lens antenna arrays [1,2], lens-array antenna [3], phased transmitarray antennas [4,5,6], rotatable transmitarrays [7,8], curved-surface transmitarrays [9], curved-surface lens [10] and so on. These works are all carried out at frequencies below 100 GHz, where the beam-scanning techniques and machining processes used therein are relatively mature. However, those quasi-optic types of beam-scanning techniques in [2,3,9,10] suffer from the difficulties of designing and fabricating the complex curved optimal focal arcs and contours of the lenses [6]. Some of the inconveniences of the scanning schemes [1,7,8] include the need to change the lens aperture or move the feed source or switch channels. We summarize four methods for beam generation in TAA and lens antennas into a comparison table with pros and cons, which is investigated in Table 1 as follows:

There are relatively few reported works realizing a high gain for frequencies higher than 100 GHz. According to the processing methods, 3-D printing lens [11], curved-surface dielectric lens [12,13], all-metallic lens or transmitarrays [14,15,16], low-temperature co-fired ceramic (LTCC) technology transmitarrays [17], and standard printed circuit board (PCB) technology transmitarrays [18,19], are adopted for high gain forming in the sub-terahertz band (100–300 GHz). A transmitarray at 140 GHz is fabricated using multilayered LTCC technology [17], where the parasitic patches are selected as transmitting elements. As reported, the receiving part is identical to the transmitting part, and the whole element is composed of six layers of substrates with intensive vias, which increases the difficulty and cost of fabrication. The transmitarray proposed in [18] realizes three-bit phase quantization at 145 GHz, based on standard printed circuit board (PCB) technology. Since the vias are also added in the most adopted unit cells, the complexity of the transmitarray is obviously increased. The one presented in [19] uses classic V-shaped transmitarray elements which scatter both polarizations at 250 GHz. This type of transmitarray element avoids vertical vias to reduce the fabrication difficulty, but reaches the limit of strip width in the fabrication. However, there is a lack of reports and studies of scanning results for high-gain antennas in the extant reported literature, and many scanning schemes for lens antennas [12,13] are still difficult to fabricate.

The array feed technique [4,5,6] is adopted to reduce the profile and cut the F/D [20]. Flexible and controllable excitation is the basis for beam diversity. The beam scanning of a conventional PAA only requires adjustment of the element excitation coefficients to achieve linear phase delay, and the excitation amplitude can be kept constant if no side lobe suppression is required. However, for array feeds, the far-field energy is a vector superposition of the secondary scattered field of the electrical large-size focusing system, rather than a direct summation of the primary radiated field of the array feed, so the determination of the excitation coefficients of the array feed is more complex. On the other hand, the linear phase shift of the PAA during beam scanning does not change the position of the array phase center significantly; whereas to achieve focal plane field matching, the array feed source must form a defocused field distribution during beam scanning, i.e., the amplitude of the array excitation must be changed so that the phase center leaves the focus of the focusing system. According to the focal plane field analysis, the excitation amplitude of different array feed units may vary greatly, which further increases the complexity of the array feed excitation coefficients. From the above comparison, it can be seen that there are many differences between array feed source and PAA in terms of design principles and methods, and determining the excitation coefficients of array feed source, i.e., beam forming algorithm is the key content in the research of array-fed transmitarray antennas.

Determining the array feed excitation coefficients at a mathematical level is an optimization-seeking process. Based on the array theory, the overall radiation pattern of the transmitarray can be regarded as a vector superposition of the radiation patterns of the aperture under the illumination of each array feed element individually. To illustrate this intuitively, this article introduces the concept of active element-lens pattern (AELP) by analogy with the active element pattern in a conventional PAA. If only the *i*th array feed element is excited and the remaining array elements are terminated with matching loads, the radiation pattern of the lens illuminated by the *i*th array feed unit is just the *i*th AELP. After obtaining the AELPs for all array feed elements by numerical calculation or full-wave calculation, the excitation coefficients of the array feed source can be determined by an optimization algorithm for a given radiation pattern requirement.

The article is devoted to developing a high-gain antenna maintaining scanning ability and filling the research gap in the sub-terahertz band. The calculation procedure is based on the array theory and avoids facing the computational pressure of full-wave calculation for electrically large-scale arrays. The radiation aperture area is limited, avoiding the replacement or extension of the TAA. The square waveguide element is adopted by this design for planar integration, thus simplifying the fabrication. A transmitarray is designed based on an element with an F/D of 0.6 at 160 GHz, and an open-ended waveguide array connected with WR-05 waveguide divider is used to excite the transmitarray. The scanning capability of the TAA is calculated, and the prototype with a 0° feeding network at 160 GHz is measured. This work highlights two innovative points. Firstly, the high-gain beams can be scanned within the limited aperture area, which solves the problem of the large propagation attenuation of the 150–170 GHz band communication system. Secondly, the full-wave calculation of large-scale arrays is avoided, and the calculated and measured results are brought into reasonable agreement by achieving the algorithm.

## 2. Fundamental Principles of Phase Compensation

In order to maintain a high-gain beam, the transmitarray aperture must provide a focused phase to compensate the propagation between the feed and the transmitarray. The focused phase provided by the *mn*th element is expressed as:(1)$$\varphi_{I,mn}=k\left|{\vec{r}}_{mn}-{\vec{r}}_{f,c}\right|,$$
where ${\vec{r}}_{f,c}$ is the position vector corresponding to the center of the feed’s aperture, and ${\vec{r}}_{mn}$ is the position vector corresponding to the *mn*th element in the transmitarray aperture. The propagation phase is generated by a fixed ideal feeding point source located at the center of the feed’s aperture, instead of located at the phase center of each port of the feed for simplicity in the calculation process. After adding the phase (1), a uniform phase distribution is obtained.

When a wave propagates through a medium, the wave path is always bent in the direction of the phase latency. Therefore, in order to manipulate the direction of the beam, an additional phase must be added to the wave launched by the array feed. The following derivation is a general case for calculating the phase of a fixed delay, i.e., the defocused phase $\varphi_{II}$. In general, let the transmitarray aperture be a surface formed by rotating an arbitrary curve around the z-axis, with the antenna elements arranged uniformly and continuously on the surface. The parametric equation versus the spherical angle $\theta_{f,c}$ of the arbitrary curve is expressed as $r(\theta_{f,c})$. As shown in Figure 1, the wave front $\bar{QR}$ of the refracted beam is in equiphase.

In order to bring the fields at points Q and R into equiphase, the phase introduced by the optical path $\bar{PQ}$ should be equal to the phase increment of the lens at point R with respect to point P, i.e., satisfying:(2)$$\frac{2\pi }{\lambda }dr+d\varphi_{II}=\frac{2\pi }{\lambda }\sin\beta \cdot dS.$$

K is the angle factor, which serves to steer the incident beam, and measures the degree of deflection of the electromagnetic wave. K is expressed as:(3)$$\beta =K\theta_{i},$$
where $\theta_{i}=\theta_{f,c}$ is the incident angle between the position vector $\vec{r}$ and the surface normal $\vec{n}$, and β is the angle between the refracted wave rays and $\vec{n}$. K does not vary with $\theta_{i}$ for simplicity in this article.

Considering the following relations:(4)$$\tan\theta_{i}=\frac{dr}{rd\theta_{f,c}},$$
(5)$$dS=\sqrt{r^{2}+{\left(\frac{dr}{d\theta_{f,c}}\right)}^{2}}\cdot d\theta_{f,c}=\frac{rd\theta_{f,c}}{\cos\theta_{i}},$$
combining (2)–(5), we get
(6)$$\frac{d\varphi_{II}}{d\theta_{f,c}}=\frac{2\pi }{\lambda }\cdot \left[r(\theta_{f,c})\cdot \frac{\sin K\theta_{f,c}}{\cos\theta_{f,c}}-\frac{dr(\theta_{f,c})}{d\theta_{f,c}}\right],$$
for planar TAA, the parametric equation of the lens can be expressed as
(7)$$r(\theta_{f,c})=\frac{H}{\cos\theta_{f,c}},$$
bringing (7) into (6), we have
(8)$$\frac{d\varphi_{II}}{d\theta_{f,c}}=\frac{2\pi H}{\lambda }\cdot \frac{\sin K\theta_{f,c}-\sin\theta_{f,c}}{\cos^{2}\theta_{f,c}}.$$

The target of scanning scope is ±5° in one dimension, so a univariate numerical integral can be performed to obtain the value of $\varphi_{II}$ across the entire aperture in the scanning direction, and the value of $\varphi_{II}$ remains the same in the non-scanning plane. By combining the focused phase and the defocused phase, the converging energy can be dispersed into the scanning scope, thus making the high-gain beams scannable.

## 3. Transmitarray Antenna Aperture Design

### 3.1. Configuration of Overall Transmitarray Antenna

The configuration of a TAA with *M* × *N* elements illuminated by a 1 × 16 array feed is shown in Figure 2, where *D* is the side length of the TAA aperture. *H* is the feeding distance from the center of the feed’s aperture to the TAA aperture along the *z*-axis. From [21], the vector element patterns $\vec{A_{mn}}$, the element excitation magnitude and phase of the *i*th port of the feed $M_{mn,i}^{feed}$, and $\varphi_{mn,i}^{feed}$, respectively, can be computed. $\vec{u}$ represents the observation direction in the far field area. To simplify the calculation, $\vec{A_{mn}}$ is replaced by the scalar approximation, where the 3 dB beam width of element radiation pattern is set as 190° [22]. The radiation patterns of the *i*th port of the feed are simulated in the Ansys high-frequency structure simulator (HFSS).

In order to obtain the initial AELPs for all array feed units in the early stage of this design, we assume that the transmitarray is ideal and completely lossless, i.e., the transmission magnitude is unity. The arrangement of actual transmitarray elements will be described in the subsequent process.

The magnitude of the *mn*th element after the introduction of the ideal transmitarray can be expressed as:(9)$$M_{mn,i}^{ideal}=M_{mn,i}^{feed}\cdot 1,$$

The phase of the *mn*th element after the introduction of the focused and defocused phases can be expressed as:(10)$$\varphi_{mn,i}^{ideal}=\varphi_{mn,i}^{feed}+\varphi_{I,mn}+\varphi_{II,mn},$$
combining (9) and (10), we have the expression for the *i*th AELP:(11)$$F_{i}(\theta ,\phi )=\sum_{m=1}^{M}\sum_{n=1}^{N}\vec{A_{mn}}(\vec{u})\cdot M_{mn,i}^{ideal}\cdot e^{j\varphi_{mn,i}^{ideal}}.$$

The amplitude and phase distribution within the aperture can be obtained by modulating the phase of the electromagnetic wave through the elements and calculating the far-field electric field value using the superposition method. After obtaining the initial AELPs, the excitation coefficients of the feed can be determined by the following optimization algorithm.

### 3.2. Transmitarray Element Design

Inspired by the waveguide element proposed in [14], a square waveguide element is employed for phase manipulation. The metal material of the waveguide array possesses several benefits, such as high mechanical strength, high power capacity, high machining accuracy and a lack of dielectric losses compared with PCB elements. The structure of the square metal waveguide element is shown in Figure 3, where *L* is the thickness along the propagation axis, $h_{x}$ and $h_{y}$ are the hollow dimensions along the *x* and *y* axis, respectively, $p_{x}$ and $p_{y}$ are the periodicities of the array along the *x* and *y* axis, respectively, and *R* represents the radius of the fillets around the corners. The dimensions $h_{x}$ and $h_{y}$ remain equal, in order to support the propagation of dual-linear polarization. The dimensions $p_{x}$ and $p_{y}$ remain equal, thus forming square lattice arrangement of the array. Fillets are necessary in mechanical processing of perforation.

The phase change Δϕ introduced by the waveguide element is calculated as follows:(12)$$\Delta \phi =\beta L=\left(k_{0}\sqrt{1-{\left(\frac{\pi }{k_{0}h_{y}}\right)}^{2}}\right)\cdot L,$$
where β is the propagation constant of the waveguide, and $k_{0}$ is the wave number in free space. The hollow dimensions ($h_{x}$ and $h_{y}$) vary from 1.1 mm to 1.4 mm. The thickness *L* is chosen as 10 mm in the early stage of design. The periodicities of the array along the *x* and *y* axis ($p_{x}$ and $p_{y}$) are both equal to 1.7 mm, which is designed to suppress the grating lobe when scanning to 5° at the upper frequency (170 GHz) according to the following condition:(13)$$p_{x}=p_{y}<d=\frac{\lambda }{1+\sin\beta_{m}},$$
where d is the threshold periodicity of the array, λ is wavelength in free space and $\beta_{m}$ is maximum scan angle, which is defined as the angle between the scan direction and the positive *z*-axis. Full-wave electromagnetic calculation of the element is carried out in HFSS, and the mutual coupling among identical elements is considered by Floquet ports and two pairs of master–slave boundary conditions along the *x* and *y* axis, respectively. Simulated transmission magnitude and phase of *L* = 10 mm element under normal incidence are shown in Figure 4.

The aperture field needs to maintain a certain transmission amplitude while correcting the phase at each point of the aperture, which is the basic requirement of the element design. The results show that the element can cover at least 360° of phase shift range in the 150–170 GHz band. However, in the subsequent step of matching the ideal design phase with the actual element at multiple frequencies, if the 10 mm thickness is selected, the compensation phase provided by the element is limited to some extent, and the total phase error is not manageable enough to be reduced by optimization. Thus, according to (12), increasing the thickness of the element will expand the phase shift range, which can be extended to exceed one period of 360°. The wider the bandwidth for the compensating phase, the richer the unit templates available in the calculation database, and thus the easier it is to reduce the total phase error when performing multi-frequency phase matching in the following section. However, considering the difficulty of the actual mechanical processing, the element cannot be thickened infinitely, and secondly, as the thickness increases, the more sensitive the machining errors versus hollow dimensions is, thereby the larger phase perturbation compensated by the actual element is. After a compromise, the thickness is taken as *L* = 16 mm, and the magnitude and phase characteristics curves at 150 GHz, 160 GHz, and 170 GHz are presented in Figure 5.

The accurate calculation of the array gain requires consideration of the oblique incidence of the electromagnetic wave to the lens, i.e., the oblique incidence case. The HFSS calculation results of the transmission performance of the elements versus the oblique incidence angle *θ* at 160 GHz are given in Figure 6.

The calculation results of element at both normal incidence and oblique incidence form the calculation database together, which will be imported into the code for the subsequent calculation process.

### 3.3. Array Feed Design

The array feed provides the illumination required for beam scanning of large-sized electrical focusing systems. The element form of the PAA is selected as the open-ended rectangular waveguide, whose specification is WR-5. The configuration of the open-ended waveguide element is shown in Figure 7.

The element is composed of one open-ended waveguide in the center and surrounding short-circuited parasitic cells, which enables the shape of radiation patterns of the element to be controllable to some extent. The calculation of the element is carried out in HFSS under 1-D periodic boundary condition along the scanning direction, which is consistent with the polarization of the electric field. The dimensions of the parasitic cells are $l_{x}$ in length, $l_{y}$ in width and $l_{z}$ in depth, and $l_{x}$ is also the diameter of the circular hole adhering to both short sides of the rectangular waveguide. After optimization, the following parameter values are determined: the thickness of metal block in *y* direction l_box = 7 mm, dimension of spacing of the elements in *y* direction $D_{y}$ = 1.5 mm, $l_{x}$ = 0.4 mm, $l_{y}$ = 1.4 mm, and $l_{z}$ = 2 mm. Simulated 2-D radiation patterns of the element are given in Figure 8.

Typically, the size of array feed source is determined by the overall scanning gain and scanning coverage of the beam scanning system. The number of elements of the phased array feed may increase beyond count and spacing of the elements may reduce to zero in order to improve beam resolution approaching infinity, ignoring the practical feasibility, but in reality, one has to make a trade-off between the beam resolution and realization of mechanical processing. As the number of ports increases, the complexity of back-end power divider increases, making the whole feeding network more difficult and costlier to process. As a result, the dimension of spacing of the elements in *x* direction $D_{x}$ is chosen as 1.2 mm, which approaches the limits of machinability of metal workpieces.

The technical goal of this article is to realize 1-D scanning; therefore, a 1 × 16 (1 row and 16 columns) linear array is formed with two open-waveguide dummy elements added to the left end and right end of the linear array as primary emitters, preparing for the high-gain beam scanning. The number of ports is selected as 16, considering the balance between beam performance and fabrication complexity of feed network. The polarization of the waveguide elements is aligned with the direction of the array. Configuration of the 1 × 16 linear array feed is given in Figure 9.

After determining the configuration of the array feed, the network at the back end is then designed. The feeding network is in the form of a metal waveguide power divider. Since there are no mature solutions for controlling the magnitude in this band in our lab, e.g., attenuators or amplifiers, the excitation magnitudes provided by the waveguide power divider are always considered equal to unity. The small inequalities in amplitude among the ports due to the conductor losses are ignored in the design process. In other words, in the following section, the array feed source provides uniform excitation magnitudes, so that to determine the phase is the only task left for the port excitation optimization.

## 4. Beam Forming Algorithm

### 4.1. Port Excitation Optimization Algorithm

The subscript of port is expressed as i, ranging from 1 to 16. For a particular spherical angle $\left(\theta_{0},\phi_{0}\right)$, when the excitation of the *i*th port is $c_{mi}\cdot e^{jc_{pi}}$, the corresponding electric field value is $F_{i}(\theta_{0},\phi_{0})\cdot c_{mi}\cdot e^{jc_{pi}}$. Among the 16 ports, the port with the largest value of the AELP is selected as the first port and determined as the highest priority port. Specify the highest priority port excitation value as $1\cdot e^{j0^{\circ }}$, i.e., $c_{m1}=1$ and $c_{p1}=0^{\circ }$. Next, superimpose the remaining 15 ports’ electric field values with that of the highest priority port in order to improve the port synthesis efficiency (PSE) as follows:(14)$$PSE_{i}=\frac{\left|\sum_{k=1}^{i}F_{k}(\theta_{0},\phi_{0})\cdot c_{mk}\cdot e^{jc_{pk}}\right|}{\sqrt{\sum_{k=1}^{i}c_{mk}{}^{2}}},$$
which is defined as the ratio of the summation of electric field value to the 2-norm of the magnitudes of port excitation. In the PSE, the numerator is the superposition value of the electric field at the far field for each port, which directly indicates the consistency of the electric field superposition, and the denominator uses the total contribution of the magnitudes input by the traversed ports as a normalization of the numerator. There is no difference in priority among the remaining 15 ports, in other words, their orders of optimization are arbitrary. Next, find the local maximum point of $PSE_{2}$,
(15)$$PSE_{2}(c_{m2},c_{p2})=\frac{\left|F_{1}(\theta_{0},\phi_{0})+F_{2}(\theta_{0},\phi_{0})\cdot c_{m2}\cdot e^{jc_{p2}}\right|}{\sqrt{c_{m1}{}^{2}+c_{m2}{}^{2}}},$$
and $\left(c_{m2},c_{p2}\right)$ can be determined. Next, find the local maximum point of $PSE_{3}$,
(16)$$PSE_{3}(c_{m3},c_{p3})=\frac{\left|\sum_{k=1}^{2}F_{k}(\theta_{0},\phi_{0})\cdot c_{mk}\cdot e^{jc_{pk}}+F_{3}(\theta_{0},\phi_{0})\cdot c_{m3}\cdot e^{jc_{p3}}\right|}{\sqrt{\sum_{k=1}^{2}c_{mk}{}^{2}+c_{m3}{}^{2}}},$$
and $\left(c_{m3},c_{p3}\right)$ can be determined. Similarly, the local maximum point of the efficiency is found step by step up to the 16th port, and $PSE_{16}$ is expressed as:(17)$$PSE_{16}(c_{m16},c_{p16})=\frac{\left|\sum_{k=1}^{15}F_{k}(\theta_{0},\phi_{0})\cdot c_{mk}\cdot e^{jc_{pk}}+F_{16}(\theta_{0},\phi_{0})\cdot c_{m16}\cdot e^{jc_{p16}}\right|}{\sqrt{\sum_{k=1}^{15}c_{mk}{}^{2}+c_{m16}{}^{2}}}.$$

So far, the excitation coefficients of all ports are optimized. 

For the *b*th spherical angle of scanning beam $\left(\theta_{b},\phi_{b}\right)$, the optimized excitation magnitudes and phases of the *i*th port at the *l*th frequency are recorded as $c_{mi,opt}(b,l)$ and $c_{pi,opt}(b,l)$, respectively. Some typical interested integer scanning angles $\theta_{b}$ are selected, −5°, −4°, −3°, −2°, −1°, 0°, 1°, 2°, 3°, 4° and 5° in the scanning range of ±5°, and for 1-D scanning, $\phi_{b}=0{}^{\circ}$, making the total number of interested beams B=11. The calculated frequency points are selected as 150 GHz, 155 GHz, 160 GHz, 165 GHz and 170 GHz in the operating band of 150–170 GHz, whose total number is L=5.

### 4.2. Fitness Functions

The first fitness function $Fitness_{1}$ is constructed to smooth the directivities while maintaining a high level in the entire operating band, as calculated by
(18)$$Fitness_{1}=BLD_{mean}+clc\cdot \frac{1}{BL}\sum_{b=1}^{B}\sum_{l=1}^{L}{\left(D(b,l)-D_{mean}\right)}^{2},$$
where D(b,l) is the directivity of the *b*th beam at the *l*th frequency, $D_{mean}$ is the mean of all the calculated D(b,l), clc is the coefficient of linear combination between the sum and variance for simplification for the following single-objective optimization. The ideal compensated phase of the lens with the combination of the focused and defocused phases is used as the optimization template, and the multi-frequency phase matching is performed by considering the effect of the actual lens elements.

The second fitness function of the optimization is the sum of the magnitude weighted phase difference of the entire TAA at each frequency point multiplied by the frequency weights, which aims at measuring the matching level between the phases compensated by actual elements and the designed template phase. The magnitude weighting is extracted from the incident field of the lens illuminated by the feed source with the optimized excitation, as calculated by
(19)$$M_{mn}^{actual}(b,l)=\left|\sum_{i=1}^{16}M_{mn,i}^{feed}(l)\cdot e^{j\varphi_{mn,i}^{feed}(l)}\cdot c_{mi,opt}(b,l)\cdot e^{jc_{pi,opt}(b,l)}\right|.$$

In order to make a balance among all the scanning beams, an average is carried out for the magnitude weighting, which is expressed as
(20)$$M_{mn}^{average}(l)=\sum_{b=1}^{B}M_{mn}^{actual}(b,l).$$

The phase error of all TAA elements at the *l*th frequency can be mathematically expressed as
(21)$$\varphi_{error}(l)=\sum_{m=1}^{M}\sum_{n=1}^{N}\left[{\left(∠S_{21}\right)}_{mn}^{element}(l)-\varphi_{I,mn}(l)-\varphi_{II,mn}(l)-\varphi_{r}(l)\right]\cdot M_{mn}^{average}(l),$$
where ${\left(S_{21}\right)}_{mn}^{element}$ is the transmission coefficient of the *mn*th element, and $\varphi_{r}(l)$ is the reference phase constant at the *l*th frequency. In [23], the reference phase constant is optimized to improve the performance of reflectarray by minimizing the total phase error at two design frequencies. Inspired by this work, the reference phase constant is involved in the optimization as the variable of the algorithm, considering the similarities of the phase compensation between reflectarray and transmitarray. The actual compensated phase is optimized to minimize the phase error; thus, the most suitable TAA element distribution is determined after traversing the whole calculation database of TAA elements.

Then, the total phase error at multiple frequencies, serving as $Fitness_{2}$, can be described as
(22)$$Fitness_{2}=\sum_{l=1}^{L}weight(l)\cdot \varphi_{error}(l),$$
where weight(l) is the weighting factor at the *l*th frequency.

### 4.3. Complete Optimization Process

The complete optimization process is summarized by the flowchart in Figure 10, where the basic particle swarm optimization (PSO) method is carried out twice in the main procedure. Two single-objective optimizations of the PSO algorithm are performed at the five frequency points of 150 GHz, 155 GHz, 160 GHz, 165 GHz and 170 GHz. In the optimization,K, clc, weight and $\varphi_{r}$ are set as the optimization variables progressively. Brief reasons for the choice of optimization variables are given here. Our goal is to combine high-gain beams with adequate steerable capability within an angular domain, so we combine the sum and the variance of all calculated directivities with linear operation. Then, an appropriate coefficient of linear combination clc is needed for a harmonization of orders of magnitude inside the single-objective fitness function. A moderate value of the angle factor K is needed to realize a relatively small scan loss on the angular field, maintaining the high-gain beams in the meantime. Since the actual element distribution is needed to ensure high performance in a frequency band simultaneously, we have to take the frequency weight into the total phase error to broaden the bandwidth of the TAA system. Equal weight selection does not smooth the gain enough, so weight is needed to be tuned by the optimization. The selection of the reference phase is to minimize the total phase error in the operating band. The reference phase constant $\varphi_{r}$ should be optimized carefully to improve the performance of TAA.

The first part of the optimization generates the ideal distribution of elements, the optimized K, clc are obtained, and the corresponding excitation coefficients are known. By executing the second part of the optimization, the optimized weight and $\varphi_{r}$ are obtained. By integrating the calculated radiation patterns, the initial TAA directivity can be obtained. The realized gain is calibrated with two factors based on the directivity, i.e., the element loss and the spillover loss. The element loss is a weighted average of the element losses used over the TAA aperture. The spillover loss is obtained by integrating the field over the TAA aperture under illumination by the feed [22].

### 4.4. Optimization Results

The optimization results are summarized in Table 2:

The calculated results of realized gain in the scanning plane and the corresponding excitation coefficients are shown in Figure 11. From Figure 11a, it can be seen that the 1-D scanning with a scope of 0° to 5° is realized at 160 GHz. The calculated optimized excitation results from 0° to 5° are given in Figure 11b, and based on the symmetry of the feed source and transmitarray elements arrangement, −5° to 0° can be achieved by simply mirroring the feed network in geometry keeping the transmitarray still. The calculated patterns and the optimized excitation of −5° to 0° are exactly the mirror of those of 0° to 5°, and are not given for brevity. The corresponding actual TAA element distribution in view of hollow dimensions consisting of 100 × 100 elements is shown in Figure 12.

## 5. Results and Discussions

### 5.1. Results and Error Analysis

To perform the verification, a typical beam direction, such as 0°, is selected during the experiment in this article, where the required excitation coefficients are provided by the waveguide power divider. Figure 13a shows the profile of the waveguide array cascaded two-dimensional waveguide feeding network. The feeding network is fabricated using a high-precision machining process. The complete structure of the two-dimensional waveguide feeding network is dissected into two parts along the symmetrical plane, so that each part of the structure can be formed by planar mechanical machining with gold-plated process. Then, the two parts are tightly assembled together using screws and positioning pins.

Note that the excitation coefficients provided by the designed 0° feeding network are the same as the optimized ones at 160 GHz, but the two do not agree at other frequencies or at other beam directions since the power divider is a single-frequency design. The simulated excitation phases provided by the designed 0° feeding network are given in Figure 13b.

A prototype of the designed TAA for verification is shown in Figure 14. The designed TAA consists of 100 × 100 elements with an F/D of 0.6. During the measurement, the 5 mm-thick absorbing material layers adhered to both surfaces of the metallic mechanical frame under the feeding illumination and the vicinity of the feed, to reduce unexpected interferences caused by reflections.

Note that the realized gain of the test result includes the conductor losses introduced by the feeding network. The measurement was performed in an anechoic chamber shown in Figure 15.

The results of the near-field measurements in the anechoic chamber are presented in Figure 16 and Figure 17.

Figure 16a shows that the measured $S_{11}$ is always below −11 dB at all operating frequencies. The measured and calculated results of the TAA are compared in Figure 16b and Figure 17. It can be seen from Figure 16b that the gain error reaches a maximum 2.2 dB at 165 GHz, which is explained in the following paragraph. Figure 17 shows the calculated Co-Pol radiation patterns and the measured ones at 150, 160, and 170 GHz, respectively, where the scanning plane corresponds to E-plane, and the non-scanning plane corresponds to H-plane. The measured Cr-Pol levels are also shown in Figure 17, which are always 25 dB below the main beam at all calculated operating frequencies.

The comparison between the test results of gain and calculation results shows that the calculation errors of 150 GHz and 160 GHz are relatively small, while at 155 GHz, 165 GHz and 170 GHz, the calculation errors are about 2 dB. The gain of the TAA system is very sensitive to the change of the compensation phase, and the accuracy of the element compensation phase directly determines the accuracy of the final calculation. The decrease in magnitude of the transmission coefficient under oblique incidence conditions will directly lead to a decrease in gain. Similarly, the phase undulation of the transmission coefficient under oblique incidence conditions will also act directly on the array gain. The calculation process requires the addition of a step to simulate the oblique incidence case. The problem encountered in the specific calculation is the accurate description of the oblique incidence data. In a specific TAA design, each element has a different incoming wave direction, and a complete, full-wave calculation of the incoming wave direction for each cell would require a very large amount of computation and correspondingly long calculation time when implemented in a large-scale array. In order to reduce the calculation time and scale, each case of local incoming wave direction is interpolated numerically. The calculation results of the compensation phase under the oblique incidence show that the compensation phases of 150 GHz and 160 GHz are almost monotonically related to the edge length of the square waveguide and the oblique incidence angle, and the interpolated compensation phases are more accurate when the interpolation is performed, yet the compensation phase curves of 155 GHz, 165 GHz and 170 GHz do not show good monotonicity, and larger interpolation errors emerge, which directly leads to larger gain calculation errors. The numerical errors brought by the interpolation are considered to explain the majority of the problem, and the machining errors of the feed source and transmitarray are also blamed for the performance deterioration, which also causes the asymmetry of the radiation pattern.

### 5.2. Comparison with Other Works

A comparison of the related publications discussed in Section 1 is provided in Table 3, where those with the operating frequencies beyond 100 GHz are focused. Works proposed in [11,13,17,19] aim at high-gain broadside beam, but the focuses of their research do not fall on scanning performance. As for beam scanning, [18] provides one broadside TAA and two steered TAA, which realize a 33.0 dBi peak gain and a maximum scanning angle of 28.7°. Meanwhile, the fly’s eye lens in [12] demonstrate multi-frequency beam scanning with a peak gain 34 dBi at 180 GHz, at the cost of complexity in curved surface processing. Finally, the scheme of epsilon-near-zero lens used in [15] achieves an 11.0 dBi peak gain and a maximum scanning angle of 18°. This work combines high-gain broadside beams exceeding 35.7 dBi with scanning capabilities, including the five calculated frequency points simultaneously by changing the optimized feeding networks, avoiding the replacement of the TAA elements.

## 6. Conclusions

In this work, a 150–170 GHz transmitarray with an all-metallic structure based on mechanical processing is proposed. The presented transmitarray is fed by an open-ended waveguide array with a F/D of 0.6. A set of defocused phases is added to the phase distribution of the monofocal transmitarray along the scanning direction, which allows the converging energy to be dispersed into the scanning scope. The measured results agree reasonably well with the calculations based on the array theory. According to the measured results, the proposed transmitarray can achieve a high gain, 37.95 dBi at 160 GHz, although a maximum of 2.2 dB error appears compared to the calculation in the operating band of 150–170 GHz. The fabrication errors and interpolation limitations are probably the reasons for the error. In summary, the proposed beam forming algorithm has been proven in the fabricated prototype to generate high-gain beams while remaining scannable in the millimeter-wave band and is expected to demonstrate its potential in other applications.

## Figures and Tables

**Figure 1 sensors-23-04709-f001:**
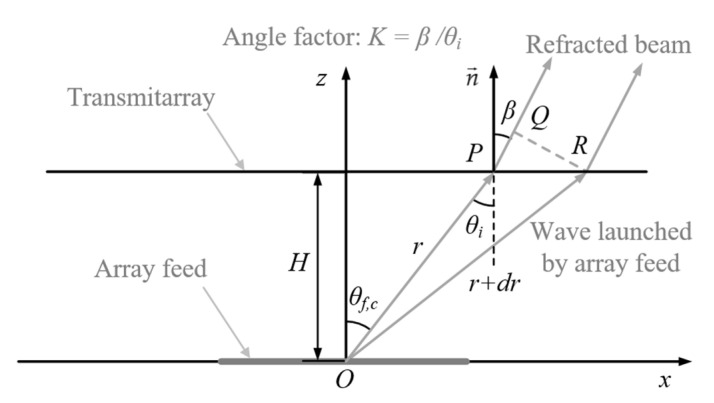
Geometry of beam refraction for generating the defocused phase.

**Figure 2 sensors-23-04709-f002:**
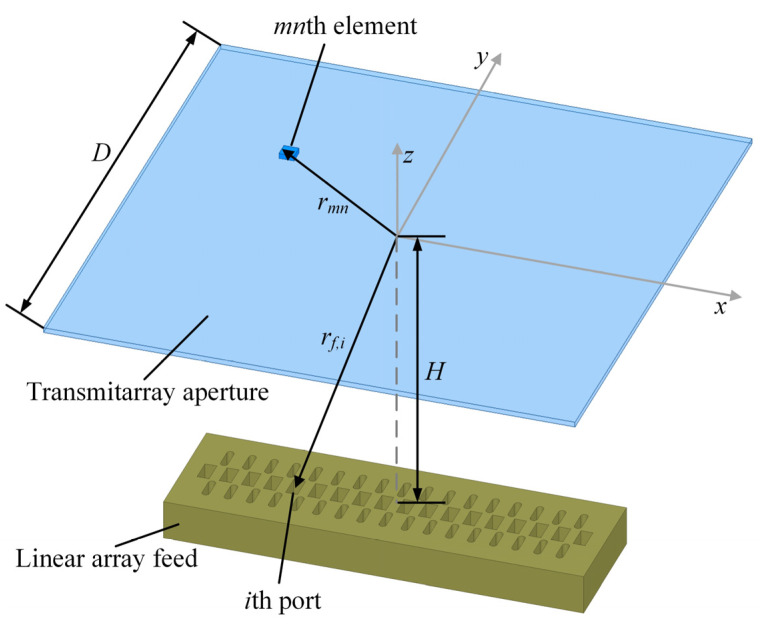
The configuration of a TAA with *M* × *N* elements illuminated by a linear array feed.

**Figure 3 sensors-23-04709-f003:**
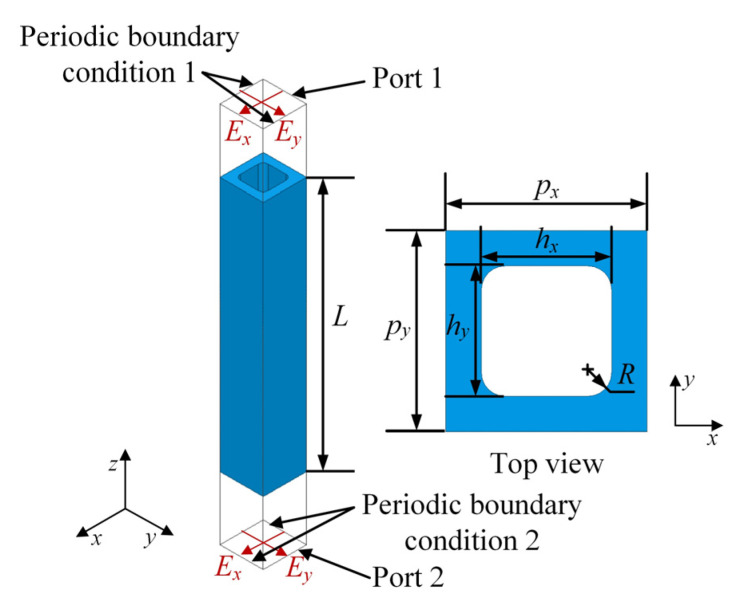
Geometry of the square waveguide element.

**Figure 4 sensors-23-04709-f004:**
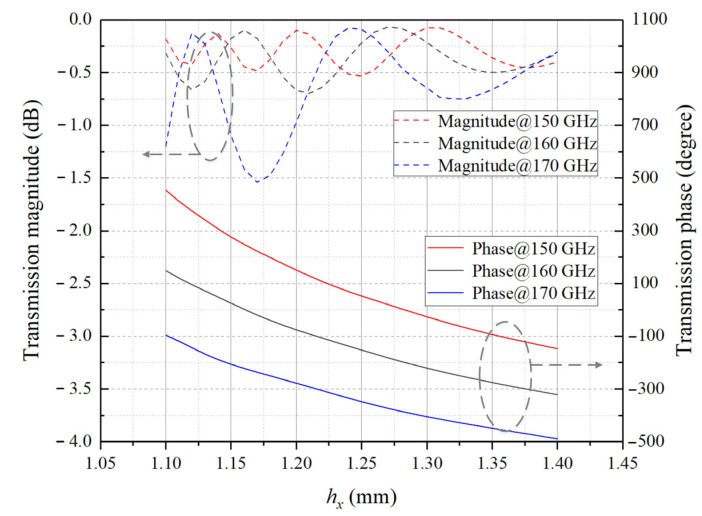
Calculation results of transmission magnitude and phase versus hollow dimension along the *x* axis of *L* = 10 mm element under normal incidence.

**Figure 5 sensors-23-04709-f005:**
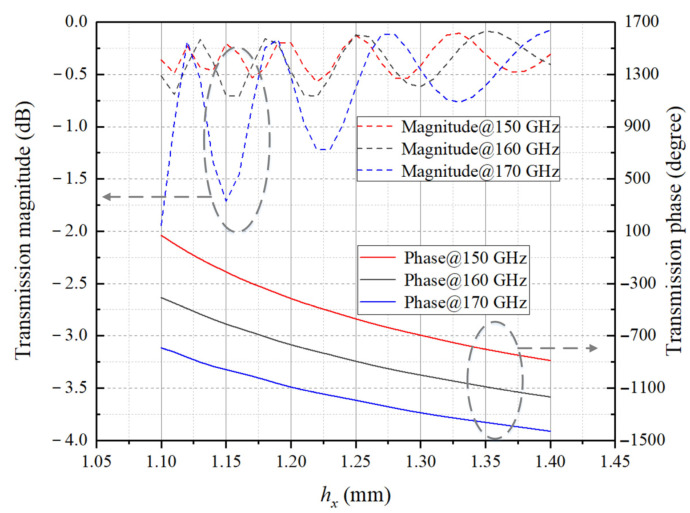
Calculation results of transmission magnitude and phase versus hollow dimension along the *x* axis of *L* = 16 mm element under normal incidence.

**Figure 6 sensors-23-04709-f006:**
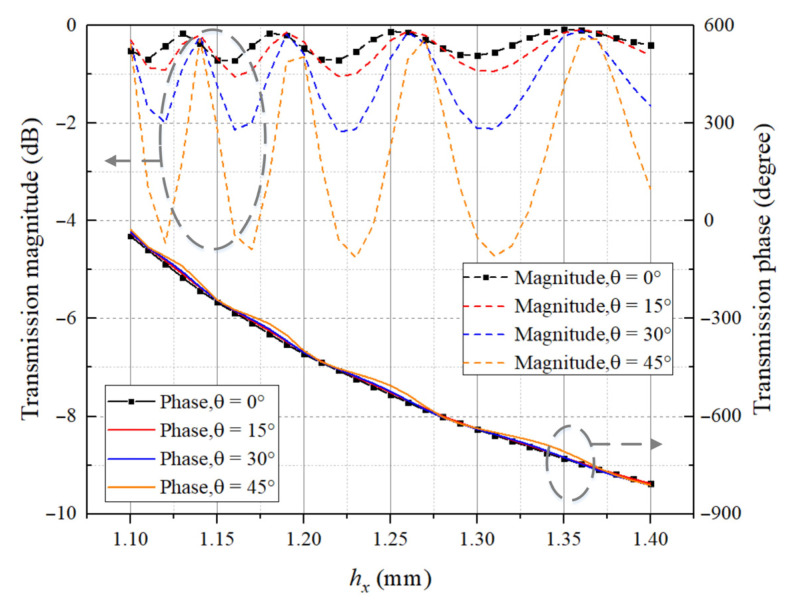
Transmission performance under several typical oblique incidences for square waveguide elements at 160 GHz.

**Figure 7 sensors-23-04709-f007:**
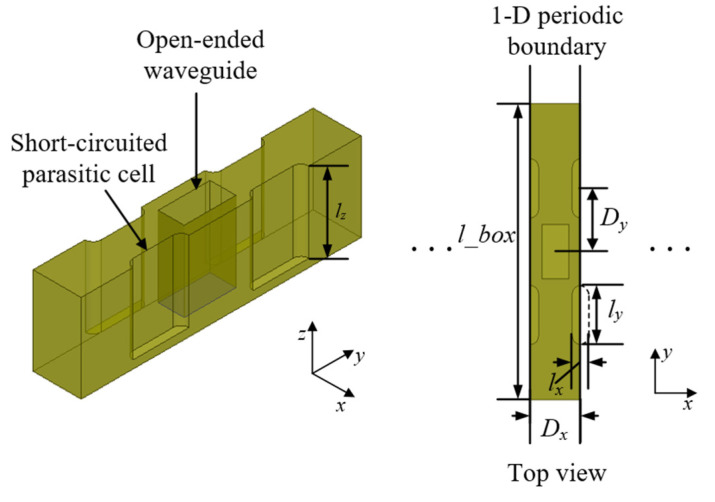
Configuration of the open-ended waveguide element.

**Figure 8 sensors-23-04709-f008:**
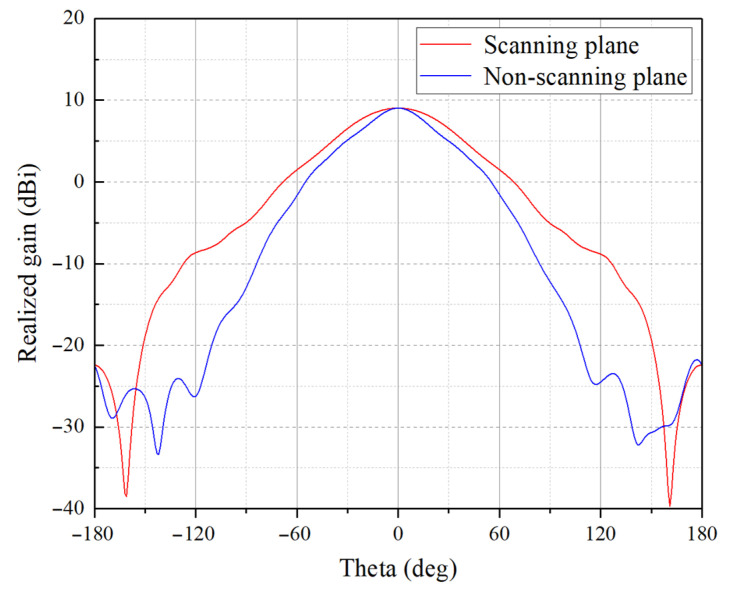
Simulated 2-D radiation patterns of the feeding element.

**Figure 9 sensors-23-04709-f009:**
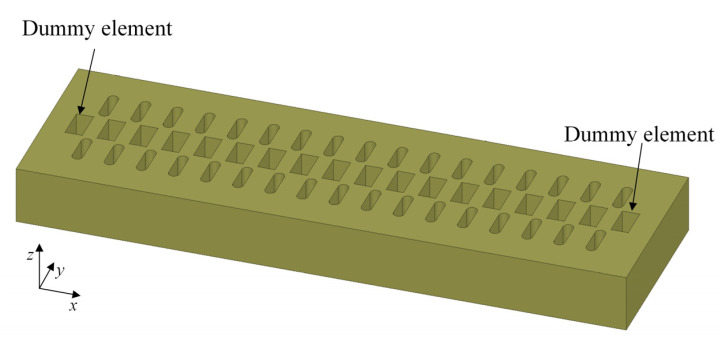
Configuration of the 1 × 16 linear array feed with two dummy elements.

**Figure 10 sensors-23-04709-f010:**
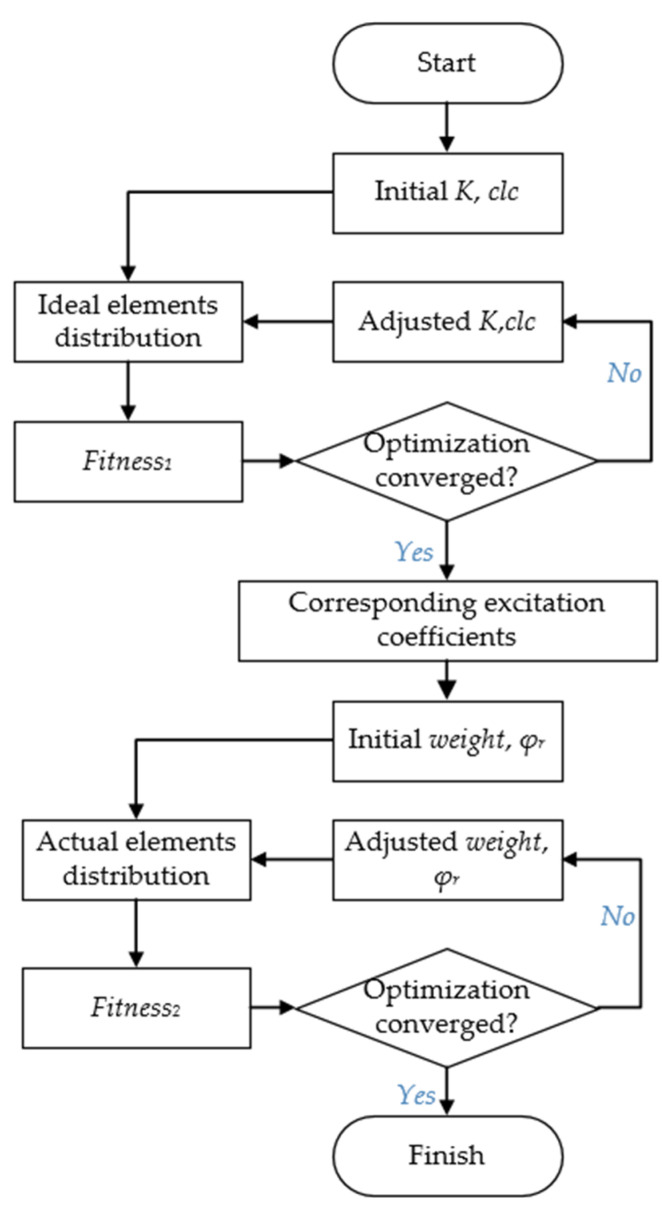
Complete optimization flowchart.

**Figure 11 sensors-23-04709-f011:**
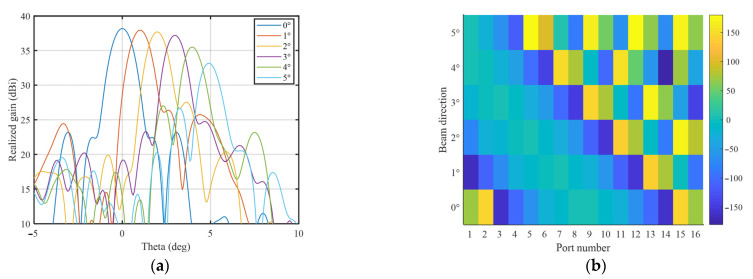
Calculated results in the scanning plane at 160 GHz: (**a**) realized gain; and (**b**) optimized excitation phases in degrees under different beam directions. The color map is used to distinguish the difference between different values of excitation phases.

**Figure 12 sensors-23-04709-f012:**
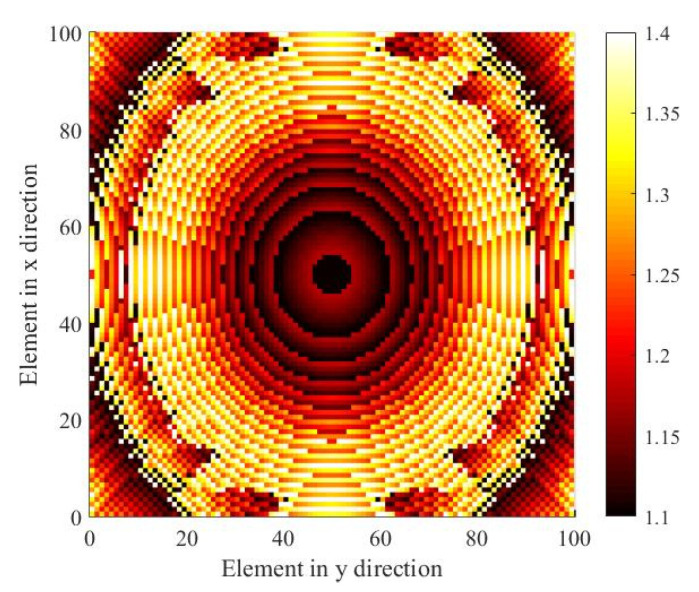
Actual TAA element distribution in mm.

**Figure 13 sensors-23-04709-f013:**
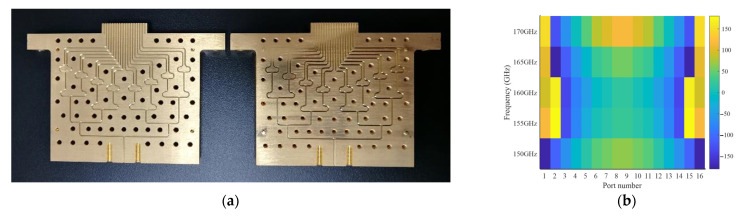
Waveguide feeding network designed at 160 GHz: (**a**) profile; and (**b**) excitation phases in degree.

**Figure 14 sensors-23-04709-f014:**
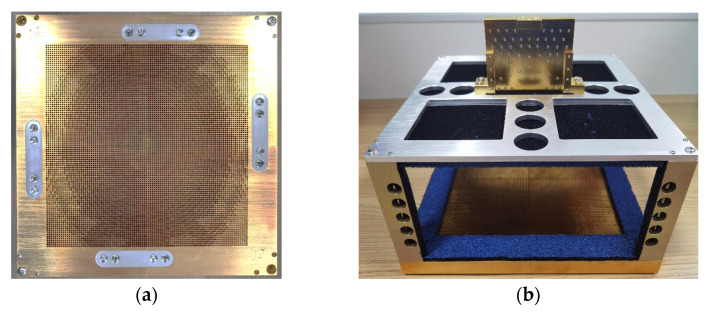
Prototype of the TAA: (**a**) aperture, and (**b**) integration with the 0° feed network.

**Figure 15 sensors-23-04709-f015:**
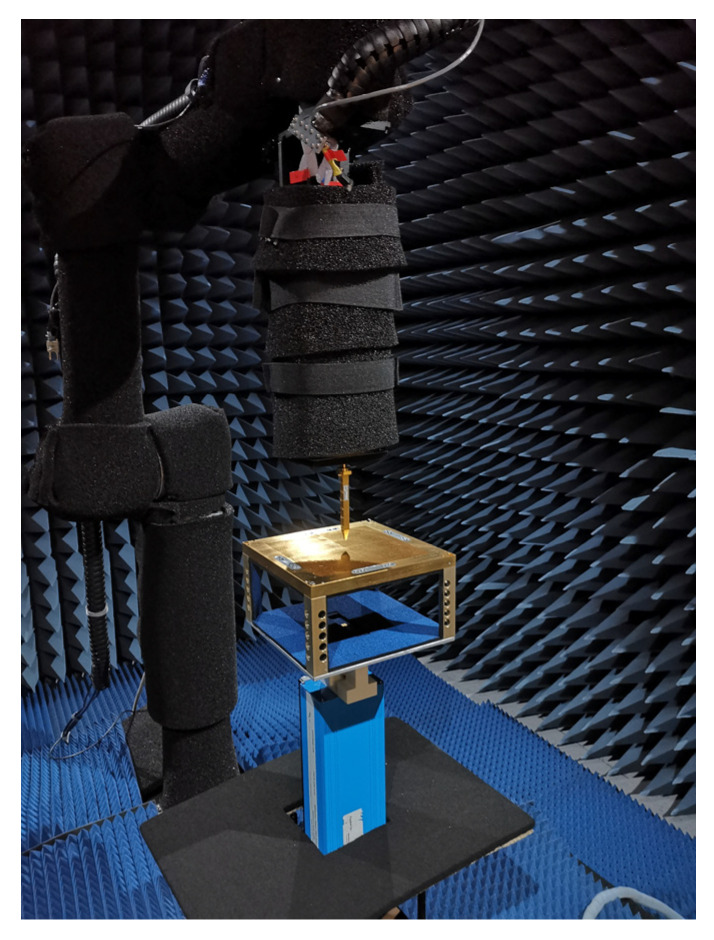
Setup for near-field measurements in the anechoic chamber.

**Figure 16 sensors-23-04709-f016:**
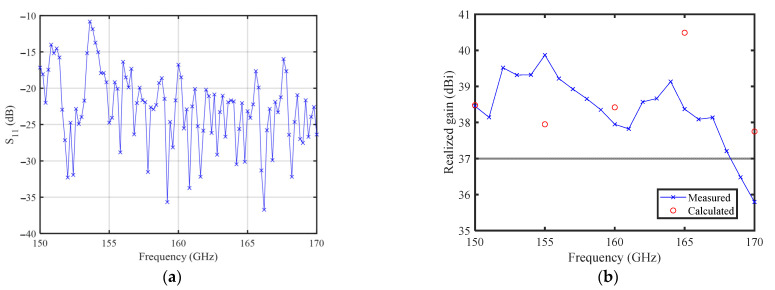
Measured parameters: (**a**) $S_{11}$ (dB); and (**b**) realized gain compared with calculated one.

**Figure 17 sensors-23-04709-f017:**
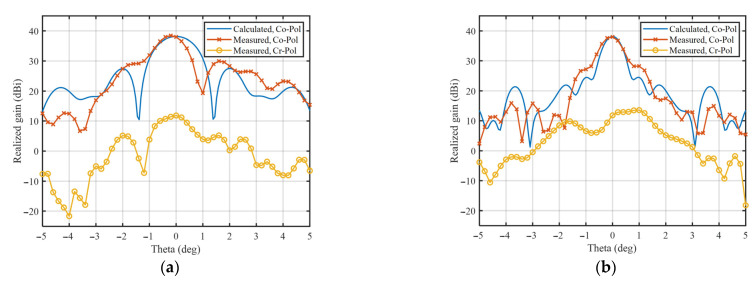
Calculated Co-Pol, measured Co-Pol and Cr-Pol radiation patterns: (**a**) 150 GHz, scanning plane; (**b**) 150 GHz, non-scanning plane; (**c**) 160 GHz, scanning plane; (**d**) 160 GHz, non-scanning plane; (**e**) 170 GHz, scanning plane; and (**f**) 170 GHz, non-scanning plane.

**Table 1 sensors-23-04709-t001:** Comparison table of typical methods of beam generation in TAA and lens antennas.

Methods	Phased Array-Fed [1,2,3,4,5,6]	Aperture Rotation [7,8]	Single-Feed Movement [9]	Multiple Channels Switching [10]
Multibeam	Yes	No	No	Yes
Continuity of scanning	Yes	Yes	Yes	No
Dimensions of scanning	1-D/2-D	2-D	1-D	1-D
Speed of scanning	Fast	Slow	Slow	Fast
Cost	High	Low	Low	Medium

**Table 2 sensors-23-04709-t002:** Optimized results under calculated frequencies.

Variables	150 GHz	155 GHz	160 GHz	165 GHz	170 GHz
K	0.81	0.81	0.81	0.81	0.81
clc	−8	−8	−8	−8	−8
weight	0.8	0.5	1	0.6	1
$\varphi_{r}$	0 deg	85 deg	218 deg	0 deg	140 deg

**Table 3 sensors-23-04709-t003:** Comparison table of this work to other works.

Reference	Frequency	Peak Gain	Steered Gain (Angle)	Scan Loss	Process
[11]	300 GHz	30.8 dBic	N/A	N/A	3-D printing
[12]	140 GHz	~31 dBi	~29 dBi (18.25°)	~2 dB	dielectric lens
	180 GHz	34 dBi	~30 dBi (17.25°)	~4 dB	
	220 GHz	~35 dBi	~32 dBi (17°)	~3 dB	
[13]	240 GHz	39.0 dBi	N/A	N/A	dielectric lens
[15]	144 GHz	11.0 dBi	7.55 dBi (18°)	3.45 dB	all metallic
[17]	140 GHz	33.45 dBi	N/A	N/A	LTCC
[18]	145 GHz	33.0 dBi	N/A	N/A (lens1)	PCB
	145 GHz	N/A	32.55 dBi (9.5°)	0.45 dB (lens2)	
	145 GHz	N/A	31.05 dBi (28.7°)	1.95 dB (lens3)	
[19]	250 GHz	28.8 dBi	N/A	N/A	PCB
This work	150 GHz	38.45 dBi (meas. ^1^)			all metallic
	155 GHz	39.87 dBi (meas.)			
	160 GHz	37.95 dBi (meas.)	33.17 dBi (5°)	5.02 dBi (calc. ^2^)	
	165 GHz	38.37 dBi (meas.)			
	170 GHz	35.79 dBi (meas.)			

^1^ Meas. means measurement, corresponding to the case of TAA beingexcited by the fabricated 0° feeding network designed at 160 GHz. ^2^ Calc. means calculation, corresponding to the case of TAA being excited by the calculated optimized excitation coefficients given in Figure 11b.

## Data Availability

Not applicable.

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
