# Peer review of "High-Gain Millimeter-Wave Beam Scanning Transmitarray Antenna"

_sensors, 2023, doi:10.3390/s23104709_

Round 1
Reviewer 1 Report
The abstract requires minor revision, the abstract did not highlight the problem statement or previous literature limitation that the authors would like to achieve in this manuscript.
The sentence "The calculation procedure is based on the array theory and avoids facing the computational pressure of full‐wave calculation for electrically large‐scale arrays", what is the significance of mentioning this in the abstract? I think this can be omitted.
In the introduction, it is recommended to include a table of comparisons, to summarize the literature review, particularly on Transmitarray Antenna, the highlights, and limitations.
Those theoretical parts from Pozar's books are well-known facts, and can be removed from the paper. Only include those being derived for the design of the Transmitarray Antenna.
In conclusion, again it is recommended to conclude how significant the finding compare to previously published work.
Minor polish on language content is required.
Reviewer 2 Report
The reasoning for the selecting optimization constants are not well developed. Please clarify.
There are some important applications of transmit arrays missing. For example, near-field transformation of medium-to-high gain electromagnetic sources is explained in the following article.
All-metal wideband metasurface for near-field transformation of medium-to-high gain electromagnetic sources, Scientific Reports 11 (1), 9421
There is some frequency shift between measured and simulated results in Fig 17 (A). Please clarify the reason
Please elaborate on the fabrication process and comment on the cost of prototyping.
Fig. 3: please add the boundary condition to Fig. 3.
N/A
